# Sparse VideoGen2: Accelerate Video Generation with Sparse Attention via Semantic-Aware Permutation

**Shuo Yang**[*]  **Haocheng Xi**[*]  **Yilong Zhao**  **Muyang Li**  **Jintao Zhang**
**Han Cai**  **Yujun Lin**  **Xiuyu Li**  **Chenfeng Xu**  **Kelly Peng**
**Jianfei Chen**  **Song Han**  **Kurt Keutzer**  **Ion Stoica**

University of California, Berkeley    MIT    NVIDIA    Stanford University

https://github.com/svg-project/Sparse-VideoGen

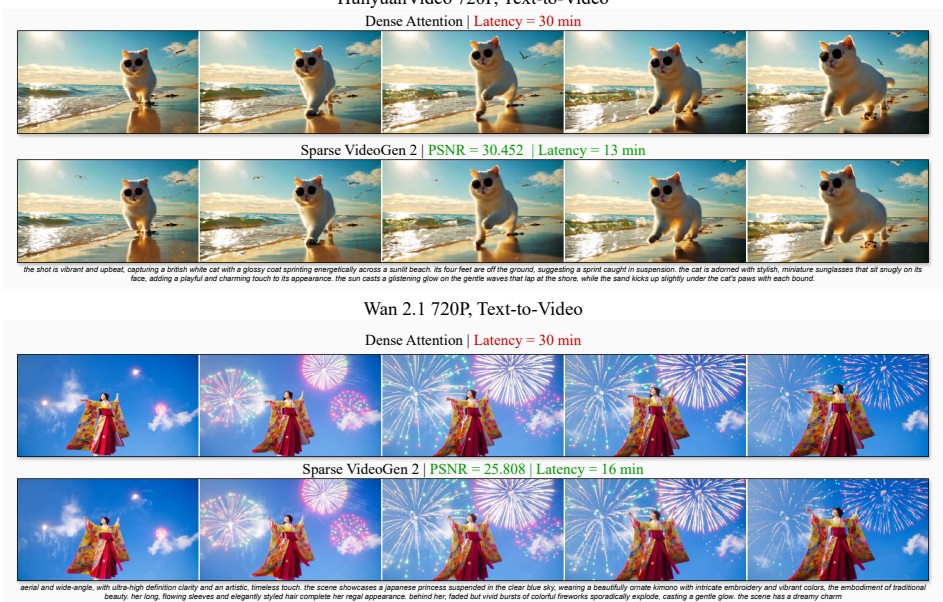

Figure 1: SVG2 accelerates video generation while maintaining high quality. On a single H100, for Hunyuan-Video and Wan 2.1, SVG2 achieves up to 2.30 and 1.89 end-to-end speedup, with a PSNR up to 30 and 26.

## Abstract

Diffusion Transformers (DiTs) are essential for video generation but suffer from significant latency due to the quadratic complexity of attention. By computing only *critical tokens*, sparse attention reduces computational costs and offers a promising acceleration approach. However, we identify that existing methods fail to approach optimal generation quality under the same computation budget for two reasons: (1) Inaccurate critical token identification: current methods cluster tokens based on *position* rather than *semantics*, leading to imprecise aggregated representations. (2) Excessive computation waste: critical tokens are scattered among non-critical ones, leading to wasted computation on GPUs, which are optimized for processing contiguous tokens. In this paper, we propose SVG2, a training-free framework that maximizes identification accuracy and minimizes computation waste, achieving a Pareto frontier trade-off between generation quality and efficiency. The core of

---

[*]Equal contribution.

39th Conference on Neural Information Processing Systems (NeurIPS 2025).

SVG2 is *semantic-aware permutation*, which clusters and reorders tokens based on semantic similarity using *k*-means. This approach ensures both a precise cluster representation, improving identification accuracy, and a densified layout of critical tokens, enabling efficient computation without padding. Additionally, SVG2 integrates top-$p$ dynamic budget control and customized kernel implementations, achieving up to $2.30\times$ and $1.89\times$ speedup while maintaining a PSNR of up to 30 and 26 on HunyuanVideo and Wan 2.1, respectively. Our code is open-sourced at https://github.com/svg-project/Sparse-VideoGen.

# 1   Introduction

Diffusion Transformers (DiTs) have demonstrated significant efficacy in generative tasks, particularly excelling in generating high-quality images and videos [1, 2, 3]. However, the computational efficiency of DiTs remains a major bottleneck, primarily due to the quadratic computational complexity introduced by 3D spatio-temporal attention mechanisms [4]. For instance, generating just a five-second video using HunyuanVideo on an NVIDIA A100 GPU takes nearly an hour, where the 3D attention accounts for more than 80% of end-to-end runtime. This inefficiency severely limits the practical deployment of DiT-based generative models.

Figure 2: Trade-off curves between generation quality (PSNR) and efficiency (density). SVG2 consistently surpasses existing methods given the same density, achieving a Pareto frontier.

To mitigate the quadratic computational complexity, previous studies have observed that self-attention mechanisms are naturally sparse, where only a small portion of computations significantly influence the final output [5, 6, 7]. Therefore, the computational costs can be dramatically reduced (up to $8\times$) with negligible degradation in generation quality by only computing the *critical tokens* [4, 8]. To effectively identify these critical tokens, existing approaches introduce an *identification step* where activations from each token are used to estimate attention scores [9, 10]. Tokens with the highest scores are then selected as critical and processed by the following sparse attention. To minimize overhead, the identification is typically processed at the *block granularity*, treating consecutive tokens as an aggregated token, which are selected or ignored as a whole [11, 12].

However, we observe that given the same computational budget (i.e., the number of selected critical tokens), existing sparse attention methods significantly fall behind the oracle generation quality, where the critical tokens are selected assuming the attention scores are known in advance rather than estimated (§ 3.2). We identify that this performance gap arises from two primary challenges:

1. Inaccurate identification: existing block-wise identification methods are ineffective in precisely identifying critical tokens. Because tokens are clustered into blocks based on *positions* rather than *semantic similarities*, tokens within the same block may have dramatically different activations in the latent space. Consequently, the aggregated activations become less representative [11], leading to inaccurate estimations of attention scores and thus incorrect identification of critical tokens. E.g., widely used techniques such as mean pooling [9] and max pooling [5] are prone to inaccuracies, particularly when applied to distinct tokens.

2. Computation waste: existing methods cause computation waste even if critical tokens could be perfectly identified. This is because of the mismatch between sparse computation and hardware specifications [13]. For instance, tensor cores on NVIDIA GPUs require a minimum matrix multiplication shape of $16 \times 16 \times 8$ [14], which necessitates a batch size of 16 tokens. Thus, even if only a subset of a block of 16 tokens are critical, the entire block must still be computed to utilize tensor cores, causing computation waste. Our empirical evaluations show that up to 80% of computation can be wasted on non-critical tokens (§ 3.2).

To bridge this gap, we propose SVG2, a training-free sparse attention approach specifically designed to accelerate video generation for DiT-based models, achieving a Pareto frontier trade-off between generation quality and computational efficiency (as shown in § 5.4). Our key insight is to leverage *semantic-aware permutation* to maximize the accuracy of critical token identification and minimize the computation waste of sparse computation. Specifically, semantic-aware permutation clusters

tokens into blocks according to the semantics of activations rather than positions. Consequently, tokens within each block exhibit closely aligned activations, ensuring more accurate aggregated representations and thereby significantly improving identification accuracy. Additionally, semantic-aware permutation *densifies* sparse computations by consolidating scattered critical tokens into compact, dense blocks. Due to their semantic similarity, tokens in a single block tend to be either all critical or all non-critical. This property ensures that computation is not wasted on blocks containing a mix of critical and non-critical tokens, thus improving computational efficiency.

To integrate semantic-aware permutation into an end-to-end framework, SVG2 introduces three key techniques. First, to implement semantic-aware permutation, SVG2 applies *k*-means clustering on the Query, Key, and Value vectors of each head and layer before the identification step. The resulting clusters are then permuted so that tokens within the same cluster are grouped together, ensuring semantically coherent blocks. Second, to enable dynamic allocation of the computational budget, SVG2 adopts a Top-*p* critical token selection strategy inspired by Tactic and Twilight [11, 15]. Specifically, SVG2 uses the centroids of clusters to approximate attention scores for each cluster, selecting tokens with the highest estimated scores until their cumulative sum reaches $p$. This approach enables dynamic budget allocation without manual adjustments. Third, to support dynamic block sizes for sparse attention, SVG2 introduces a customized kernel implementation. This is essential because the clusters formed by semantic-aware permutation naturally vary in size, and existing block sparse attention kernels, which are limited to fixed block sizes, cannot efficiently handle such variability.

We prototype SVG2 based on an open-sourced video generation framework [4] and customize kernels with FlashInfer [16]. We evaluate SVG2's quality and efficiency on representative video generative models including HunyuanVideo [1] and Wan 2.1 [2]. Results demonstrate that SVG2 consistently achieves a Pareto frontier, delivering superior generation quality at any given computational budget. Specifically, SVG2 delivers significant efficiency improvements, achieving an end-to-end speedup of up to $2.30\times$ and $1.84\times$ speedup while maintaining high visual quality with a PSNR of up to 30 and 26 on HunyuanVideo and Wan2.1-I2V, outperforming all prior methods.

## 2 Related Work

**Sparse Attention for Video DiTs.** Sparse attention mechanisms for accelerating DiTs fall into two categories: *static* and *dynamic*, depending on whether to select critical tokens dynamically during runtime or statically offline. Static methods [8, 4] predefine sparse patterns offline, such as identifying recent tokens as critical [4]. These methods lack adaptability to diverse sparsity patterns, leading to suboptimal performance. Dynamic methods [9, 10, 17, 18, 19, 20, 21, 22] determine sparse patterns at runtime, selecting critical tokens through an additional identification step. However, existing dynamic methods fail to achieve both high identification accuracy and low computation waste. In contrast, SVG2 consistently achieves superior performance under the same computation budget.

**Sparse Attention for Large Language Models (LLMs).** Sparse attention for LLMs falls into two categories: *memory-efficient* and *compute-efficient*. Memory-efficient methods [6, 5, 7, 23, 24] reduce memory load to accelerate decoding but are ineffective for compute-bound DiT-based video generation. Compute-efficient methods [25, 26, 27, 12, 28, 29] focus on processing only critical tokens, yet cannot directly optimize video DiTs due to unique sparse patterns of video data. Notably, MMInference [12] introduces a modality-aware permutation for multi-modal LLMs. This permutation is rule-based, designed to permute inter-modality tokens.

**Linear Attention for Diffusion Models** Linear attention [30, 31, 32] and state space models [33, 34] have gained attention in video generation, where the long-context problem makes attention module dominate the latency. Matten [35] uses the mamba block to capture global information and uses attention for local information. LinGen [36] adopts a combination of Mamba2 and Swin attention block for 1-minute video generation. M4V [37] proposes an MM-DiM block to overcome the adjacency of mamba in capturing complex spatial-temporal dynamics. Block-wise SSM scanning scheme [38] achieves long-term memory ability in video generation. TTT [39] proposes to use RNNs and designs a new TTT block for minute-long video generation. SANA [40, 41, 42] and DC-AE [43, 44, 45] reduce the generation overhead by using Linear Attention and Deep-Compressed Auto-Encoders. This line of research reduces the complexity of the attention from quadratic to linear, making it efficient in long video generation.

**Long Video Generation and Caching-Based Acceleration** Minute-level long video generation poses new challenges in the field of video generation in both quality and efficiency. CausVid [46] and Self-Forcing [47] distill a bidirectional diffusion model to an autoregressive one to leverage the efficiency of KV cache. LongLive [48] achieves multi-prompt minute-level video generation with KV cache refreshing. Framepack [49] compressed the token number within frames to reduce the sequence length. RifleX [50] and Freelong [51] extend the video generation length by context extrapolation in a training-free manner. RadialAttention [52], VMOBA [53], Mixture-of-Context [54], and VSA [55] adopt sparse attention in long-context fine-tuning to reduce the training overhead. Caching-based methods [56, 57, 58, 59, 60] optimized the efficiency by utilizing the redundancy between timesteps and classifier-free guidance (CFG). These lines of research is orthogonal to our sparse attention method, and can be integrated to achieve higher speedup.

## 3 Motivation

### 3.1 Attention in DiTs is Inherently Sparse

**Attention operation in DiTs is costly.** During each denoising step, DiTs transform the input activations with hidden dimension $d$ into Query ($Q$), Key ($K$), and Value ($V$) tensors, followed by a self-attention operation to produce the final output $O$ [61]:

$$O = P \times V, \quad P = \texttt{softmax}\left(\frac{QK^\top}{\sqrt{d}}, \texttt{dim} = -1\right)$$

where the *attention score* $P$ captures the relationship among tokens However, computing $P$ has a quadratic complexity relative to the sequence length. State-of-the-art DiTs typically process thousands of tokens per frame across multiple frames, creating a significant performance bottleneck. E.g., generating a 33-frame video using HunyuanVideo-T2V-13B requires over $80\%$ of the total end-to-end time to be spent on the attention alone [4, 8].

**Attention operation in DiTs is highly sparse.** Fortunately, attention is inherently sparse, where only a small subset of computations significantly contributes to the final output. This sparsity arises from the characteristics of the $\texttt{softmax}$ function, where a few largest values in $Q \times K^\top$ dominate the attention score $P$, which in turn dictates the final weighted output $P \times V$ [6, 11].

To quantitatively assess this sparsity, we collect attention maps from Wan2.1-I2V-14B video generation and visualize the average *recall* of attention scores under varying computational budgets, defined by the number of critical $K$ tokens. Specifically, the critical tokens are selected following an *oracle policy*, where tokens are ranked in descending order based on their attention scores. This approach illustrates the upper bound of achievable recall under a constrained computational budget.

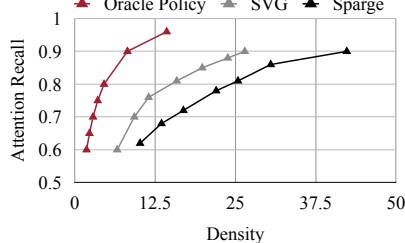

Figure 3: Comparison of attention recall versus density (i.e., number of sparse computation normalized by total computation) for the oracle policy, SVG, and SpargeAttention. Notably, the significant gap between the oracle policy and existing methods highlights the potential for improvement.

As depicted in Figure 3, the attention is highly sparse, where only $13\%$ of the computations (i.e., the percentage of attention map retained in the sparse attention) are sufficient to achieve an attention recall of $95\%$, maintaining a near-lossless PSNR of 27 while providing up to $2\times$ theoretical end-to-end speedup. This observation highlights an opportunity to leverage the trade-off between generation quality and computational efficiency.

### 3.2 Existing Sparse Attention Fails to Match the Oracle Policy

Despite the potential of sparsity in reducing computational cost, directly adopting the oracle policy is impractical. This is because identifying critical tokens requires calculating the full attention scores $P$ by $Q \times K$, thus providing no actual speedup. To achieve practical efficiency, state-of-the-art approaches [9, 10] implement a coarse-grained identification strategy. Specifically, they cluster *consecutive tokens* into large blocks and calculate attention scores at the block level, providing an

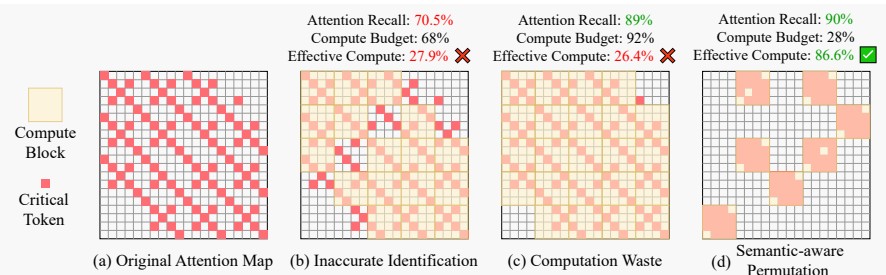

Figure 4: Illustration of how existing methods fall short due to the *inaccurate identification* and *computation waste*, assuming a computation unit of $4 \times 4$ block. (a) Original attention map of a demonstration example. (b) Position-based clustering groups distinct tokens within the same clustering, causing the imprecise representation of mean-pooling or max-pooling. Therefore, blocks with smaller number of critical tokens are ignored, causing lower recall of attention scores. (c) Due to the scattered layout of critical tokens, even if achieving a high attention recall, each compute block processes both critical and non-critical tokens, thus causing computation waste and decreasing *effective compute* on critical tokens. (d) Semantic-aware permutation clusters and reorders similar tokens into contiguous layout, thus achieving high attention recall while minimizing computation waste.

approximation of the original $P$. This approach significantly reduces the identification overhead, with less than 1% computation compared to the full attention when using a block size of $128$.

However, existing coarse-grained approaches reduce identification accuracy and lead to computation waste. As illustrated in Figure 3, existing sparse attention mechanisms consistently fall significantly short of achieving the attention recall of the oracle policy, regardless of the computation budget. This performance gap arises primarily from two key factors:

**Position-based clustering leads to inaccurate identification.** Existing methods reduce identification overhead by clustering consecutive tokens into blocks. For instance, SpargeAttention [9] groups every $128$ query tokens and $64$ key tokens, using mean pooling to create a single representation for each block, which is then used to approximate $P$. However, this position-based clustering does not guarantee semantic similarity among tokens. Tokens within the same block can exhibit vastly different activations in the latent space. For example, two physically close objects in a video frame, such as an apple and a cake, may have no semantic relationship. This variability within a block degrades the quality of the aggregated block-wise representation, leading to reduced identification accuracy. We illustrate this issue in Figure 4 and provide a quantitative analysis of the identification accuracy in § 5.5. To address this problem, we propose using semantic-aware clustering instead of position-based clustering, as detailed in § 4.1.

**Scattered critical tokens cause computation waste.** Even if all critical tokens are perfectly identified, existing sparse attention mechanisms cannot achieve the theoretical computational savings promised by the oracle policy due to a mismatch between scattered sparse computations and hardware specifications. Since the criticality of tokens is determined by semantics, critical tokens are naturally scattered across the tensor rather than being contiguous. However, modern ML accelerators, such as NVIDIA GPU tensor cores, are optimized for dense matrix multiplication, which requires contiguous input dimensions [14, 62]. As a result, scattered critical tokens must be padded with non-critical tokens to maintain a contiguous layout, leading to significant computation waste. This issue is visualized in Figure 4, and we further quantify the computation waste in § 5.5. To approach the performance of the oracle policy, an automatic permutation is required to rearrange scattered critical tokens into a dense layout to minimize computation waste, as detailed in § 4.1.

## 4 Methodology

In this section, we introduce SVG2, a training-free sparse attention framework designed to use *semantic-aware permutation* to achieve a Pareto frontier trade-off between the generation quality and computational efficiency for video DiTs. We visualize the workflow of SVG2 in Figure 5. At the core of SVG2 is semantic-aware permutation, which aims to maximize identification accuracy of critical tokens and minimize computation waste (§ 4.1). To dynamic select and adjust the computation budget, SVG2 proposes centroid-based top-$p$ selection, which enables practical deployment (§ 4.2).

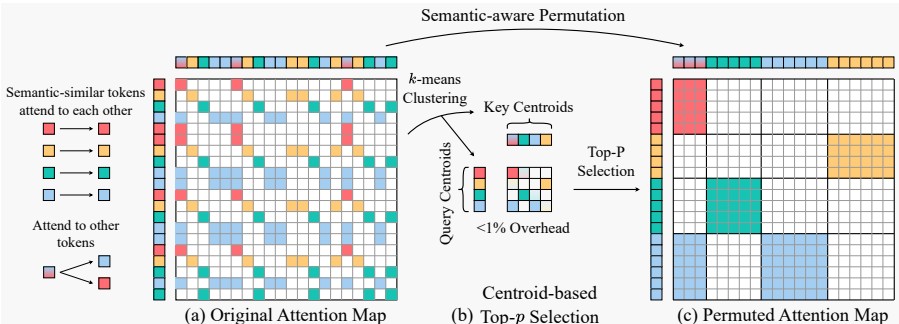

Figure 5: Overview of SVG2.[†] (a) Original attention map of a demonstration example, with various colors representing various semantics. Only tokens with similar semantics attend to each other, having high attention scores thus selected as critical tokens. (b) After $k$-means clustering, semantic-similar tokens (i.e., similar colors) are grouped into the same cluster, with the query and key centroids to precisely represent the cluster-level semantics. These centroids are then used to approximate the attention score for accurate identification of critical tokens. (c) Combined with Top-$p$ selection, critical tokens can be dynamically identified in a contiguous layout.

Additionally, SVG2 investigates several system-algorithm co-designs, such as fast $k$-means and attention kernel, significantly accelerating video generation (§ 4.3).

## 4.1 Semantic-Aware Permutation with $k$-means Clustering

As discussed in § 3.2, existing sparse attention mechanisms suffer from inaccurate identification due to the position-based clustering. To this end, SVG2 proposes using semantic similarity to cluster rather than position, by performing $k$-means on the activations of the input tokens.

Specifically, for each attention head and transformer layer, $k$-means is independently applied to query tokens ($Q \in \mathbb{R}^{N_q \times d}$) and key tokens ($K \in \mathbb{R}^{N_k \times d}$), where $N_q$ and $N_k$ represent the number of tokens in $Q$ and $K$, creating $C_q$ query clusters $Q_1, \ldots, Q_{C_q}$ and $C_k$ key clusters $K_1, \ldots, K_{C_k}$. This approach enables tokens within each cluster to share similar semantics, improving the precision of centroid representation for better identification, as detailed in § 4.2.

Furthermore, to densify the sparse computation of scattered critical tokens, SVG2 performs semantic-aware permutation based on the $k$-means clustering. Although the semantically similar tokens are logically clustered, they are physically scattered in the tensors, resulting in substantial computational waste as described in § 3.2. To address this, SVG2 permutes tokens within each cluster into a contiguous layout. Such cluster-wise contiguous layout can be efficiently computed by the underlying ML accelerators, thus reducing computation waste. We detail the permutation algorithm and the mathematical equivalence for the attention output in the following formulations. Assuming $\pi_q \in \mathbb{R}^{N_q \times N_q}$ and $\pi_k \in \mathbb{R}^{N_k \times N_k}$ be the permutation matrices such that $\pi_q \pi_q^\top = I$ and $\pi_k \pi_k^\top = I$, the permuted tokens are then $Q' = \pi_q Q$, $K' = \pi_k K$, and $V' = \pi_k V$, where $K$ and $V$ share the same permutation $\pi_k$ to guarantee the output equivalence. The permuted attention output $O'$ is:

$$O' = \pi_q^\top \text{Attention}(Q', K', V') = \pi_q^\top \text{softmax}\left(\frac{(\pi_q Q)(\pi_k K)^\top}{\sqrt{d}}\right) \pi_k V$$

$$= (\pi_q^\top \pi_q) \text{softmax}\left(\frac{QK^\top}{\sqrt{d}}\right)(\pi_k^\top \pi_k)V = \text{softmax}\left(\frac{QK^\top}{\sqrt{d}}\right)V = O$$

## 4.2 Centroid-Based Top-$p$ Selection

Despite the semantic-coherent clusters provided by the semantic-aware permutation, it remains impractical to deploy SVG2 without addressing two critical challenges: (1) how to effectively estimate the criticality of clusters, and (2) how to dynamically determine the number of selected critical clusters (i.e., the number of critical tokens) to satisfy arbitrary accuracy requirements.

---

[†]Cross-category centroids arise from SVG2's independent clustering, which enables a many-to-many map where multiple query clusters share important key clusters.

**Accurate and efficient estimation of criticality.** To estimate the criticality of each cluster, SVG2 introduces a centroid-based estimation of attention scores $P$. Specifically, it approximates the criticality of each token by estimating its $P$ using the centroids of its cluster, mimicking the oracle policy defined in § 3.1. As formulated in Equation 1, this approach calculates the pre-softmax scores $S$ using the centroids of each cluster. These scores are then weighted by the number of tokens within the cluster (i.e., the size of the cluster), to generate an approximate attention score $P'$ in Equation 2, providing an estimation of the actual $P$.

$$S_{ij} = \frac{\texttt{centroid}(Q_i) \cdot \texttt{centroid}(K_j)^T}{\sqrt{d_k}} \quad (1) \qquad\qquad P'_{ij} = \frac{|K_j| \exp(S_{ij})}{\sum_{k=1}^{C_k} |K_k| \exp(S_{ik})} \quad (2)$$

Since tokens within the same cluster already share similar semantics, the centroids can serve as highly accurate representations of the actual activations, ensuring the reliability of such estimation. Furthermore, because the typical number of clusters (i.e., $C_q$ and $C_k$) is less than 1024, the computational overhead for this approximation is negligible compared to the full attention calculation, typically accounting for less than 1% of the total computational cost.

**Dynamic adjustment of computation budget.** To dynamically adjust the number of critical tokens instead of pre-defined constant, SVG2 employs a Top-$p$ selection strategy based on the approximated $P'$. SVG2 first sorts all potential clusters in descending order according to their corresponding $P'$. It then selects clusters sequentially until the accumulated $P'$ reaches a predefined target.

### 4.3    Efficient System-Algorithm Co-design

**Fast $k$-means with centroid cache.** While $k$-means clustering is essential to semantic-aware permutation, its iterative process can introduce substantial latency if the number of iterations is large before convergence. For example, with the state-of-the-art GPU implementation of $k$-means++ [63], it takes more than 100 iterations to converge, consuming 50% or even comparable time to the attention computation. Fortunately, DiTs are known to be similar between consecutive denoising steps [59, 64], enabling reusing the centroids from the previous step as the fast initialization for $k$-means in the next step. Based on this observation, SVG2 implements a *centroids cache*, which automatically caches and reuses centroids between consecutive steps. This technique reduces the runtime of $k$-means by up to 76×, as evaluated in § 5.3.

**Efficient sparse attention kernel for varied block-sizes.** While existing efficient attention implementations (e.g., FlashAttention [65], FlexAttention [66], and FlashInfer [16]) support block-wise sparse computation, they only support a static block size (e.g., 128 × 128). However, the sizes of clusters after semantic-aware permutation are naturally dynamic and diverse, causing computation waste with the static block size. For example, SVG2 could generate a query cluster with 128 tokens with a key cluster with 32 tokens. Such 128 × 32 computation needs to be padded into 128 × 128 to use existing kernels, which causes 75% computation waste. To address this, SVG2 implements a customized attention kernel that accepts dynamic block-sizes as input.

Our dynamic block-sparse attention kernel supports both FA2 (A100) and FA3 (H100), combining sparse loading and dense computation. For FA3, we use wgmma (m64n64k16) for dense compute to maximize hardware efficiency. For query tokens, we load contiguous tokens from the same cluster, which are naturally contiguous in memory after permutation. For key/value tokens, which may be scattered in global memory due to varying cluster sizes, SAPAttn uses per-token address offsets to perform sparse loading and stores them in shared memory in a contiguous layout. This enables efficient use of MMA instructions without the need for expensive key/value padding, leading to over 85% of the theoretical maximum performance, where the upper bound is estimated by multiplying the sparsity density with the runtime of the dense FlashAttention-3. We evaluate the kernel efficiency in § 5.3.

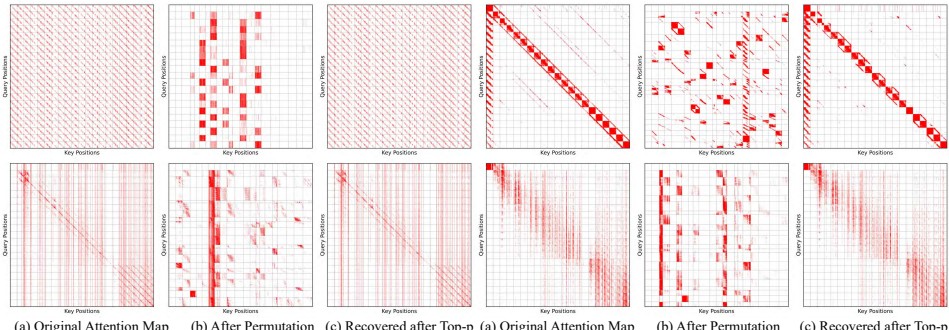

| (a) Original Attention Map | (b) After Permutation | (c) Recovered after Top-p | (a) Original Attention Map | (b) After Permutation | (c) Recovered after Top-p |

Figure 6: Visualization of attention maps from different attention heads in Wan2.1 when generating videos from VBench [67]. (a) Original attention maps with diverse sparse patterns, assuming critical tokens highlighted in red. (b) Permuted attention maps. After semantic-aware permutation, critical tokens are permuted into a contiguous layout based on the *k*-means clustering, enabling efficient block-wise computation without waste. (c) Recovered attention maps after applying centroid-based top-*p* selection and undoing the permutation. The high similarity between the original and recovered attention maps demonstrates the effectiveness of SVG2.

# 5 Experiment

## 5.1 Setup

**Models.** We evaluate SVG2 on open-sourced state-of-the-art video generation DiT models including Wan2.1-I2V/T2V-14B [2], and HunyuanVideo-T2V-13B [1] to generate videos with 720p resolution. After being tokenized by 3D-VAE, Wan2.1 generates 21 frames with 3600 tokens per frame, while HunyuanVideo processes 33 frames with 3600 tokens per frame.

**Metrics.** We assess the similarity of generated video compared to full attention using the following metrics: Peak Signal-to-Noise Ratio (PSNR), Learned Perceptual Image Patch Similarity (LPIPS), and Structural Similarity Index Measure (SSIM). We use VBench [67] to evaluate the video quality. To quantify the efficiency of sparse attention mechanisms (i.e., computational budget), we use *density*, which is defined as the sparse attention computation divided by the full attention computation. To assess end-to-end efficiency, we use the total amount of computation (i.e., FLOPs) needed for generating videos.

**Datasets.** For text-to-video generation, we adopt the prompt in Penguin Benchmark after prompt optimization provided by VBench team. For image-to-video generation, we adopt the prompt-image pairs provided by VBench [67] and crop images to 16 : 9 ratios for 720p resolution.

**Baselines.** We compare SVG2 against state-of-the-art sparse attention algorithms including static method Sparse VideoGen (SVG) [4], and dynamic methods SpargeAttention [9] and XAttention [10]. Note that we skip the evaluation of XAttention on Wan2.1 as it is not supported yet. For SVG, SpargeAttention, and XAttention, we use their official configurations.

**Implementations.** We prototype SVG2 as an end-to-end framework with customized kernels from FlashInfer [16] and benchmark on NVIDIA H100 GPU with CUDA 12.8. For SVG2, we choose $C_q = 100$ and $C_k = 500$, and explain the choice in § D. To showcase the trade-off between generation quality and efficiency, we evaluate on various accuracy target (i.e., attention score recall) as detailed in § 5.4. We also sample a single data point for detailed comparison as shown in Table 1. We conduct experiments with sparse attention skipped during the first 30% of denoising steps for all methods, as these steps are critical for generation quality. following previous work [64, 68, 56, 59]. For experiment results without warmup, please check Table 2 in Appendix.

## 5.2 Quality Evaluation

We first qualitatively showcase the effectiveness of our proposed method by showing the visualization of attention maps. As shown in Figure 6, we collect attention maps from different attention heads when running Wan2.1 on prompts from VBench. Despite the diversity of the sparse patterns (i.e., different columns in Figure 6), semantic-aware permutation effectively densifies critical tokens into contiguous layout, which enables efficient computation without waste. Furthermore, by applying

Table 1: Quality and efficiency benchmarking results of SVG2 and baselines. Warmup steps is set to 30%.

| Model | Config | PSNR ↑ | SSIM ↑ | LPIPS ↓ | VBench ↑ | Density ↓ | FLOP ↓↑ | Speedup ↑ |
|---|---|---|---|---|---|---|---|---|
| **Wan 2.1** | *14B, 720P, Image-to-Video* | - | - | - | 0.841 | 100% | 526.76 PFLOPs | 1× |
| | SpargeAttn | 21.181 | 0.665 | 0.333 | - | 38.99% | 366.80 PFLOPs | 1.47× |
| | SVG | 24.059 | 0.813 | 0.174 | 0.836 | 30.25% | 343.88 PFLOPs | 1.56× |
| | Ours | 26.562 | 0.861 | 0.138 | 0.838 | 31.28% | 346.59 PFLOPs | 1.58× |
| | Ours-Turbo | 24.510 | 0.812 | 0.179 | 0.836 | 14.13% | 301.62 PFLOPs | 1.84× |
| **Wan 2.1** | *14B, 720P, Text-to-Video* | - | - | - | 0.846 | 100% | 658.46 PFLOPs | 1× |
| | SpargeAttn | 20.519 | 0.623 | 0.343 | 0.820 | 42.03% | 468.46 PFLOPs | 1.44× |
| | SVG | 22.989 | 0.785 | 0.199 | 0.837 | 30.25% | 429.86 PFLOPs | 1.58× |
| | Ours | 25.808 | 0.854 | 0.138 | 0.842 | 29.51% | 427.43 PFLOPs | 1.60× |
| | Ours-Turbo | 23.682 | 0.789 | 0.196 | 0.838 | 12.87% | 372.89 PFLOPs | 1.89× |
| **Hunyuan** | *13B, 720P, Text-to-Video* | - | - | - | 0.850 | 100% | 612.38 PFLOPs | 1× |
| | SpargeAttn | 27.892 | 0.884 | 0.151 | - | 42.62% | 399.16 PFLOPs | 1.53× |
| | XAttention | 28.892 | 0.898 | 0.120 | 0.839 | 39.32% | 386.90 PFLOPs | 1.56× |
| | SVG | 29.157 | 0.905 | 0.120 | 0.845 | 29.86% | 351.75 PFLOPs | 1.91× |
| | SVG + FP8 | 29.033 | 0.902 | 0.121 | 0.843 | 29.86% | 351.75 PFLOPs | 2.3× |
| | Ours | 30.452 | 0.910 | 0.117 | 0.852 | 25.45% | 335.36 PFLOPs | 2.30× |
| | Ours + FP8 | 30.389 | 0.908 | 0.118 | 0.851 | 25.45% | 335.36 PFLOPs | 2.55× |

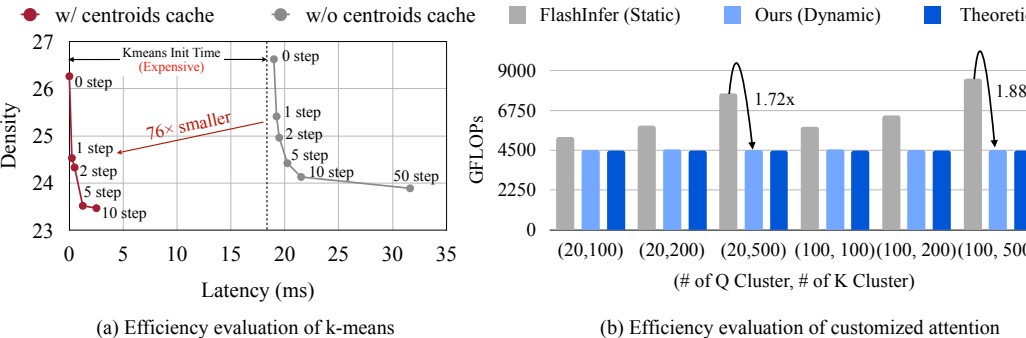

(a) Efficiency evaluation of k-means

(b) Efficiency evaluation of customized attention

Figure 7: Efficiency evaluation for fast *k*-means with centroids cache and customized attention kernel.

centroid-based top-*p* selection and then undoing the permutation, the permuted attention map is recovered into the original layout, which shows high similarity to the original attention map.

To quantitatively assess the generation quality, we evaluate the quality of videos generated by SVG2, when compared to baselines and report the results in Table 1. SVG2 consistently outperforms all baseline methods in terms of PSNR, SSIM, and LPIPS, while still maintaining the highest speedup. Specifically, SVG2 achieves an average PSNR of 26.5 on Wan2.1 and 30.4 on HunyuanVideo, demonstrating its effectiveness on generating highly consistent and smooth videos.

Due to space limitations, the full results of VBench can be found at Table 3 and Table 4 in the appendix.

### 5.3 Efficiency Evaluation

**Efficient *k*-means with centroids cache.** To demonstrate the effectiveness of centroids cache in improving efficiency of *k*-means, we compare the density achieved by SVG2 to reach the 90% attention recall, when varying different number of execution time (i.e., number of *k*-means iterations). We use widely-used algorithm *k*-means++ [63]. Since the *k*-means quality directly determine the accuracy of critical token identification, it also determines the achieved density. The lower the density is, the better the *k*-means is. As shown in Figure 7(a), enabling centroids cache reduces the end-to-end latency of *k*-means by 76× when reaching comparable or lower density. This demonstrates the effectiveness of centroids cache, which greatly reduce the initialization time.

**Efficient attention kernel with dynamic block-sizes.** To showcase the efficiency of our customized attention kernels with dynamic block-sizes, we evaluate the computation FLOPs of our implementa-

tion compared with a state-of-the-art attention library, FlashInfer [16]. We vary different combination of hyper-parameters (i.e., number of clusters $C_q$ and $C_k$) and apply centroid-based top-$p$ selection to generate the practical workloads of dynamic block sizes. We fix the attention recall as 90%. As shown in Figure 7(b), our customized kernels achieve an average of $1.48\times$ computation reduction. On practical setup with $C_q = 100$, $C_k = 500$, ours achieves $1.88\times$ reduction of computation waste.

**End-to-end speedup evaluation.** To showcase the end-to-end speedup of SVG2, we incorporate several efficiency metrics, including density, FLOPs, and end-to-end speedup into Table 1. SVG2 achieves an average speedup of $1.82\times$ while maintaining the highest generation quality. We also include a SVG2-Turbo to showcase the efficiency potential, which maintains a similar generation quality as baselines but achieves much higher speedup. Specifically, SVG2-Turbo achieves $2.5\times$ smaller density compared to SVG while achieving an even better PSNR of 23.7. Such results can be cross-validated with the sensitivity evaluation in § 5.4.

## 5.4 Sensitivity Test on Quality-Efficiency Trade-off

To validate the effectiveness of SVG2, we conduct a comprehensive evaluation on Wan2.1-I2V-14B, comparing it against baseline methods across a wide range of computational budgets (i.e., density). As shown in Figure 2, SVG2 consistently achieves better generation quality at any given density, positioning it on the Pareto frontier of the quality-efficiency trade-off. Notably, SVG2 reduces density by up to $2.3\times$ while maintaining the same PSNR.

## 5.5 Ablation Study on Semantic-Aware Permutation

**Effectiveness on improving identification accuracy.** To assess the effectiveness of semantic-aware permutation in improving the accuracy of critical token identification, we measure attention recall by comparing semantic-aware permutation-enabled and semantic-aware permutation-disabled configurations across varying computational budgets. Both methods use mean-pooling and maintain the same cluster size for consistency. As shown in Figure 8, semantic-aware permutation consistently achieves higher attention recall, indicating more accurate identification of critical tokens. This improvement is attributed to the semantic-coherent clusters generated by semantic-aware permutation, which offer precise representations.

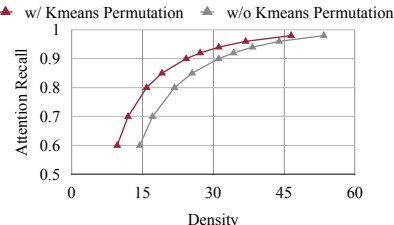

Figure 8: Attention recall across various densities. Enabling permutation consistently surpasses disabling permutation.

**Effectiveness on reducing computation waste.** We further investigate the impact of semantic-aware permutation on reducing computational waste. Specifically, we use the same set of critical tokens selected by centroid-based top-$p$ selection. For the semantic-aware permutation-enabled configuration, we feed the contiguous layout generated after permutation into GPUs, while for the semantic-aware permutation-disabled configuration, we use the scattered layout before permutation. As shown in our results, enabling semantic-aware permutation reduces computational overhead by an average of 36%, while maintaining the same set of critical tokens.

## 6 Conclusion & Limitation

In this paper, we proposed SVG2, a training-free sparse attention approach for accelerating DiT-based video generation. By clustering tokens based on semantic similarity, SVG2 accurately identifies critical tokens. By permuting critical tokens into a contiguous layout, SVG2 effectively reduces the computation waste. Comprehensive evaluations show that SVG2 achieves a superior trade-off between generation quality and efficiency, making video generation more efficient and practical. The major limitation of this paper lies in the lack of discussion and evaluation on whether the proposed methods can be extended to attention mechanisms other than DiTs.

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

# A  Visualization of the Generated Videos

We provide visualization comparison between SVG2 and Dense Attention on HunyuanVideo and Wan 2.1. Results in Figure 9 and Figure 10 demonstrate that SVG2 can preserve high pixel-level fidelity, achieving similar generation quality compared with the dense attention. Real video samples are provided in the supplementary materials.

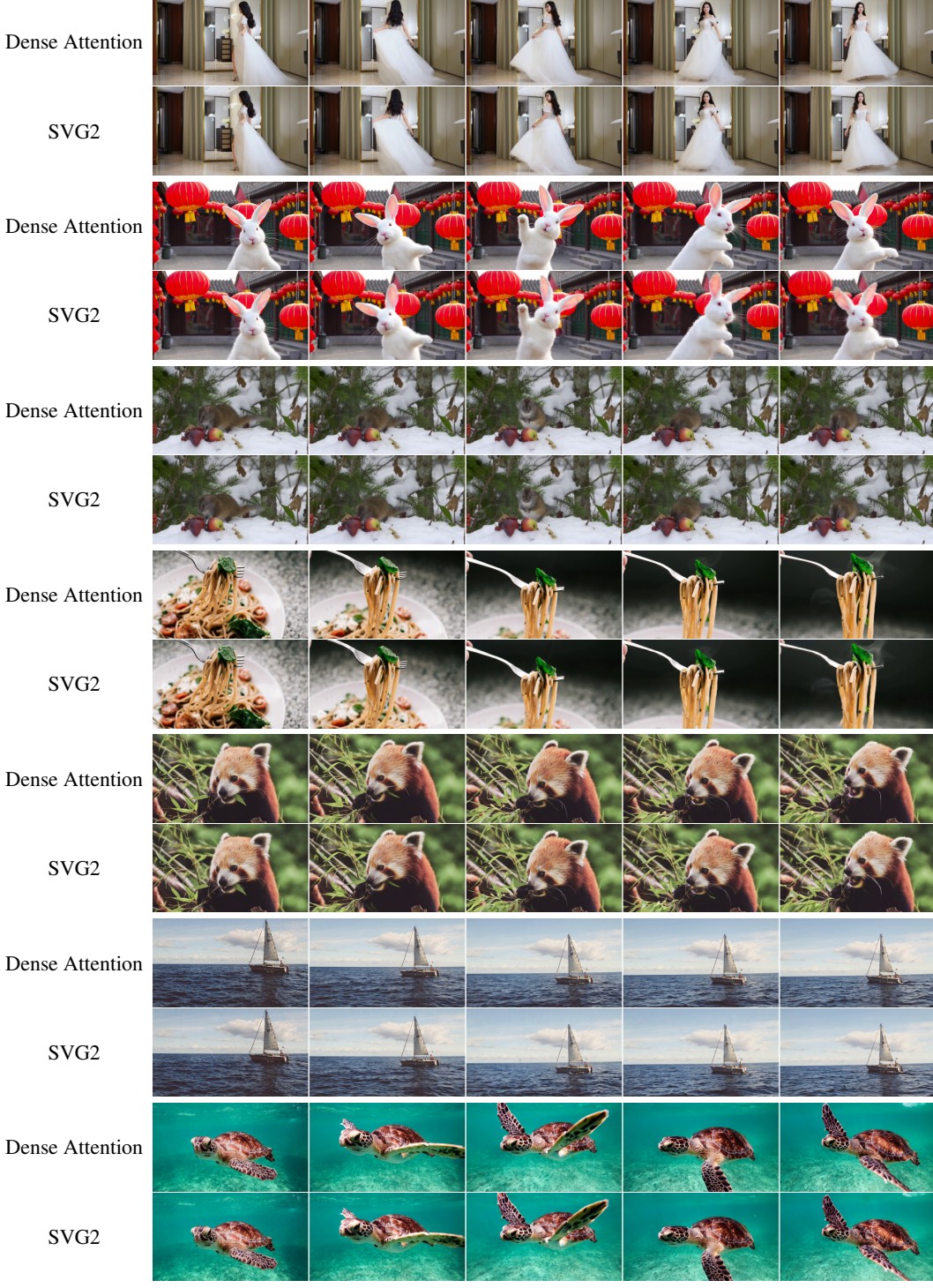

Figure 9: Comparion of Dense Attention and SVG2 on HunyuanVideo and Wan 2.1 Text-to-Video generation.

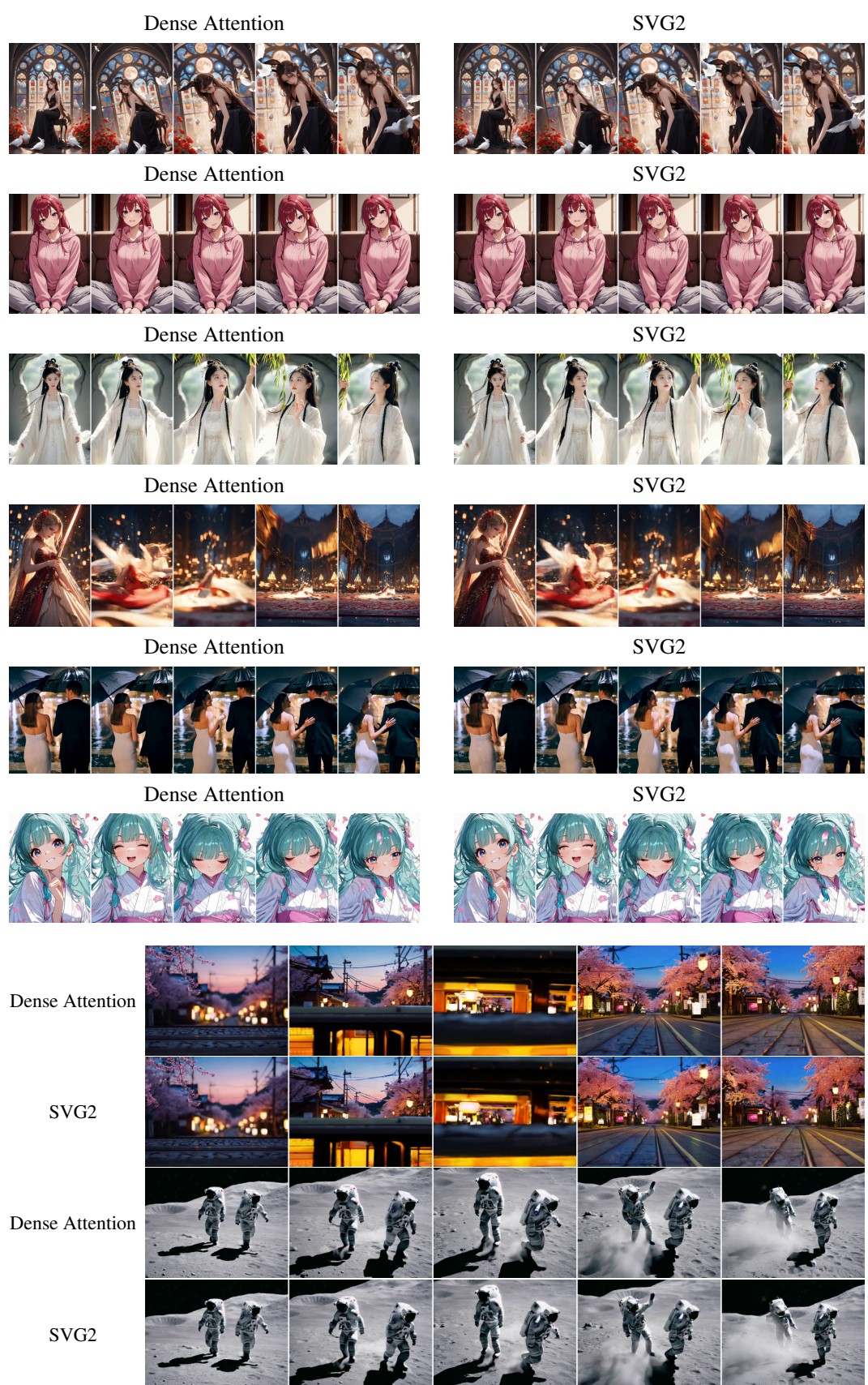

Figure 10: Comparison of Dense Attention and SVG2 on Wan 2.1 Image-to-Video generation.

# B  Performance Comparison in Warmup-free Setting

We present the performance comparison between SVG2 and the baseline without warmup steps. We find that our method consistently offers better quality under a warmup-free setting.

Table 2: Quality and efficiency benchmarking results of SVG2 and baselines. Warmup steps is set to 0%.

| Model | Config | PSNR ↑ | SSIM ↑ | LPIPS ↓ | VBench ↑ | Density ↓ | FLOP ↓ | Attn Speedup ↑ | Speedup ↑ |
|---|---|---|---|---|---|---|---|---|---|
| **Wan 2.1** | *14B, 720P, Image-to-Video* | - | - | - | 0.841 | 100% | 526.76 PFLOPs | 1× | 1× |
| | SVG | 15.608 | 0.512 | 0.404 | 0.823 | 29.54% | 262.85 PFLOPs | 2.26× | 1.86× |
| | Ours | 18.276 | 0.615 | 0.317 | 0.832 | 29.34% | 262.10 PFLOPs | 2.95× | 2.10× |
| **Wan 2.1** | *14B, 720P, Text-to-Video* | - | - | - | 0.851 | 100% | 658.46 PFLOPs | 1× | 1× |
| | SVG | 13.294 | 0.407 | 0.512 | 0.849 | 29.54% | 328.56 PFLOPs | 2.28× | 1.89× |
| | Ours | 16.502 | 0.562 | 0.373 | 0.852 | 30.12% | 331.28 PFLOPs | 2.98× | 2.13× |
| **Hunyuan** | *13B, 720P, Text-to-Video* | - | - | - | 0.820 | 100% | 612.38 PFLOPs | 1× | 1× |
| | SVG | 12.298 | 0.492 | 0.483 | 0.808 | 29.86% | 240.05 PFLOPs | 3.45× | 2.48× |
| | Ours | 19.879 | 0.735 | 0.260 | 0.816 | 28.94% | 235.16 PFLOPs | 4.06× | 2.69× |

# C  VBench Results

We provide the full VBench results of SVG2 and baselines in Table 3 and Table 4. These results clearly shows that SVG2 outperforms all other baselines.

Table 3: VBench result of SVG2. Warmup steps is 0%.

| Model | Config | SubConsis | BackConsis | MotionSmooth | AesQual | ImagQual | Average |
|---|---|---|---|---|---|---|---|
| **Wan 2.1** | *14B, 720P, Image-to-Video* | 0.946 | 0.956 | 0.979 | 0.618 | 0.709 | 0.841 |
| | SVG | 0.916 | 0.935 | 0.976 | 0.591 | 0.698 | 0.823 |
| | Ours | 0.936 | 0.946 | 0.977 | 0.597 | 0.700 | 0.832 |
| **Wan 2.1** | *14B, 720P, Text-to-Video* | 0.970 | 0.970 | 0.992 | 0.612 | 0.708 | 0.851 |
| | SVG | 0.963 | 0.969 | 0.991 | 0.612 | 0.708 | 0.849 |
| | Ours | 0.971 | 0.970 | 0.992 | 0.624 | 0.707 | 0.852 |
| **Hunyuan** | *13B, 720P, Text-to-Video* | 0.888 | 0.938 | 0.994 | 0.594 | 0.685 | 0.820 |
| | SVG | 0.867 | 0.930 | 0.991 | 0.594 | 0.656 | 0.808 |
| | Ours | 0.888 | 0.935 | 0.994 | 0.589 | 0.675 | 0.816 |

Table 4: VBench result of SVG2. Warmup steps is 30%.

| Model | Config | SubConsis | BackConsis | MotionSmooth | AesQual | ImagQual | Average |
|---|---|---|---|---|---|---|---|
| **Wan 2.1** | *14B, 720P, Image-to-Video* | 0.946 | 0.956 | 0.979 | 0.618 | 0.709 | 0.841 |
| | SVG | 0.941 | 0.948 | 0.978 | 0.606 | 0.709 | 0.836 |
| | Ours | 0.943 | 0.951 | 0.977 | 0.606 | 0.709 | 0.838 |
| **Wan 2.1** | *14B, 720P, Text-to-Video* | 0.956 | 0.968 | 0.983 | 0.613 | 0.713 | 0.846 |
| | SpargeAttn | 0.927 | 0.948 | 0.978 | 0.567 | 0.684 | 0.820 |
| | SVG | 0.947 | 0.960 | 0.980 | 0.597 | 0.703 | 0.837 |
| | Ours | 0.954 | 0.965 | 0.982 | 0.602 | 0.709 | 0.842 |
| **Hunyuan** | *13B, 720P, Text-to-Video* | 0.915 | 0.941 | 0.993 | 0.648 | 0.753 | 0.850 |
| | XAttention | 0.912 | 0.924 | 0.992 | 0.631 | 0.739 | 0.839 |
| | SVG | 0.914 | 0.928 | 0.993 | 0.652 | 0.739 | 0.845 |
| | Ours | 0.917 | 0.946 | 0.993 | 0.657 | 0.751 | 0.852 |

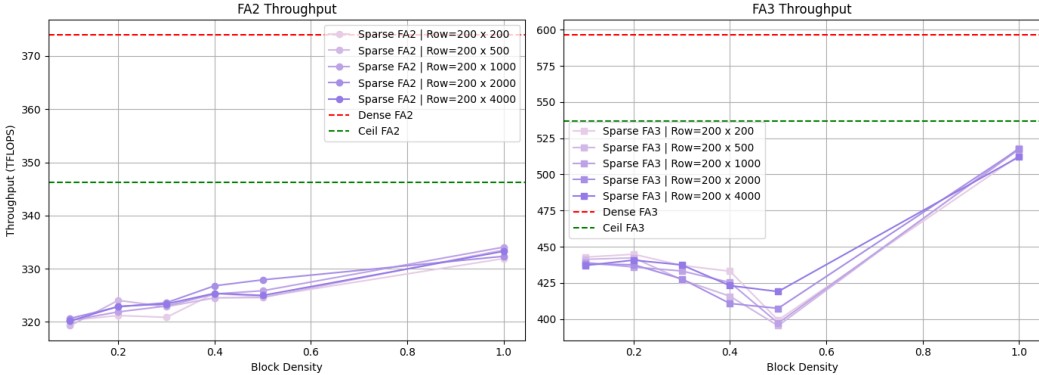

Figure 11: Efficiency evaluation for our attention kernel. We fix the number of query clusters and vary the number of key clusters.

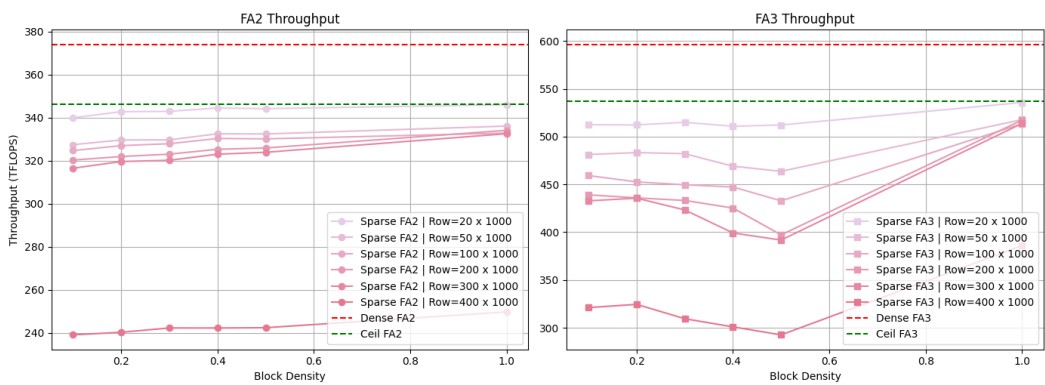

Figure 12: Efficiency evaluation of our attention kernel, where we fix the number of key clusters and vary the number of query clusters.

# D Ablation on the Number of Clusters

## D.1 Effect on the Sparse Attention Kernel

Our sparse attention kernel is fully compatible with both FlashAttention-2 and FlashAttention-3. We implemented Sparse VideoGen2 on both FA2 and FA3 backends, achieving substantial speedups on A100 and H100 GPUs. Under the Wan 2.1 setting with a sequence length of 74,256 on H100, we benchmark kernel performance by comparing sparse and dense attention across varying densities and cluster configurations. In each experiment, we control either $C_q$ or $C_k$, fixing one while varying the other. Results in Figure 11 and Figure 12 demonstrate that the kernel's performance will drastically decrease when $C_q$ is larger than 200. However, the performance is nearly identical as we increase $C_k$ to as large as 4000. This suggests us to adopt a larger $C_k$ than $C_q$.

## D.2 End-to-end Latency-quality Trade-off

We further varied the Q/K cluster counts and measured both PSNR and end-to-end efficiency. Our results in Table 5 show that setting $Q = 100$ and $K = 500$ provides the best balance between generation quality and efficiency. While increasing the number of clusters generally improves quality, efficiency can degrade due to hardware layout constraints. In particular, tensor cores require fixed input sizes (e.g., 64 for m64n64k16) to fully utilize computation, meaning that each Q cluster

| $C_q$ | $C_k$ | PSNR | SSIM | LPIPS | Speedup |
|-------|-------|--------|-------|-------|---------|
| 100 | 250 | 25.497 | 0.801 | 0.182 | 1.90x |
| 100 | 1000 | 26.276 | 0.825 | 0.159 | 1.71x |
| 50 | 500 | 22.561 | 0.742 | 0.258 | 1.90x |
| 200 | 500 | 26.213 | 0.820 | 0.157 | 1.78x |
| 400 | 500 | 26.488 | 0.868 | 0.132 | 1.25x |
| 100 | 500 | 26.128 | 0.816 | 0.169 | 1.89x |

Table 5: Performance comparison across different QC and KC settings.

must contain at least 64 tokens on average. Cluster counts beyond $Q = 100$ or $K = 500$ lead to underutilization, reducing efficiency despite potential quality gains.

### D.3 Ablation on Permutation

We performed ablation studies to investigate whether the query and key representations can adopt the same clustering strategy. Specifically, we evaluated three variants: applying Q clustering permutation $\pi_Q$ to both Q and K, applying K clustering permutation $\pi_K$ to both, and using clustering based on hidden states before QKV linear layer (i.e., shared QK embedding) for both sides $\pi_S$. As shown in Table 6, all three variants led to worse PSNR even with more computation budget (i.e., density) compared to clustering Q and K independently.

Table 6: Permutation comparison with corresponding Density and PSNR values

| Permutation used by Q | Permutation used by K | Density | PSNR |
|:---:|:---:|:---:|:---:|
| $\pi_Q$ | $\pi_K$ | 31.28% | 26.562 |
| $\pi_Q$ | $\pi_Q$ | 38.23% | 22.439 |
| $\pi_K$ | $\pi_K$ | 38.58% | 22.183 |
| $\pi_S$ | $\pi_S$ | 87.27% | 26.495 |

To further understand this, we compared the permutations of Q and K clustering and found that the permutation patterns differ substantially. Specifically, we calculate the Adjusted Rand Index value between Q clusters and K clusters, and the average ARI is 0.345, which is not very high. Therefore, clustering Q and K independently is necessary for preserving the expressiveness of attention.

## E   Performance Gap between HunyuanVideo and Wan 2.1

### E.1   Quality Difference

In our experiments, we find that the quality performance (e.g., PSNR, SSIM, LPIPS) on Wan 2.1 is generally lower than HunyuanVideo across all methods. The reason is that Hunyuan is relatively robust against precision variance while Wan2.1 is highly sensitive. For instance, when evaluating the same dense attention using different backends (FlexAttention, FlashAttention, Torch SDPA), Wan2.1 exhibited PSNR as low as 27–28. However, HunyuanVideo exhibits 33-34 PSNR despite no setup changes. Therefore, it is natural that SVG2 achieves a lower PSNR on Wan compared with HunyuanVideo, due to its sensitivity to numerical changes. These differences are largely model-specific and reflect varying sensitivity to low-level numerical behaviors, which do not correlate with the performance of the methodology.

### E.2   Speedup Difference

We also find that the speedup result on Wan 2.1 is generally lower than HunyuanVideo ($1.89\times$ versus $2.30\times$). The difference in end-to-end speedup between HunyuanVideo and Wan primarily stems from their varying attention cost ratios, which are mainly due to different context lengths and model architectures. Specifically, HunyuanVideo's context length is 118k, while Wanx's context length is 75k. HunyuanVideo has 2 parts in its layers: Self Attention and Feed-Forward Network, while Wanx

has an additional cross-attention block. Therefore, the attention proportion in HunyuanVideo will be larger than WanX. Since our method primarily accelerates the attention module via SVG2, the overall speedup naturally scales with its contribution to total runtime. We will revise Table 1 to separate attention-level and end-to-end speedups to improve clarity explicitly.

