# OpenReview forum: "Sparse VideoGen2: Accelerate Video Generation with  Sparse Attention via Semantic-Aware Permutation"
_NeurIPS.cc/2025/Conference — NeurIPS 2025 spotlight_

### Official Review · Reviewer_8YJY · 2025-06-23

**Clarity:** 3
**Significance:** 3
**Originality:** 3
**Rating:** 5
**Confidence:** 4

**Summary:**

Authors propose a method "SAPAttn" for accelerating inference speed of DiTs. The idea is quite simple: rearrange Q/K into smaller groups via k-means, and compute the attention with only the group mass. Video generation experiments are done with Wan2.1 and HunyuanVideo, with acceleration ratio of 1.89~2.30 without obvious quality deterioration.

**Questions:**

1) How much gain can we get if the proposed method is applied to DiT based image generation model? This is not discussed in the manuscript.
2) The contribution of the work lies mostly in its custom implementation of "sparse attention", will the code be released? what's the schedule?

**Ethical Concerns:**

["NO or VERY MINOR ethics concerns only"]

**Final Justification:**

All my questions are addressed properly by the authors, I'd recommend to include the result on image generation task in the final manuscript. I raised my final score to 5.

**Limitations:**

yes

**Quality:**

3

**Strengths And Weaknesses:**

Most parts of the manuscript are clearly written. The idea is not hard to follow, novelty is good as respected to the standard of NeurIPS, experiments and results are also solid. Overall it's a pretty good work. Some minor concerns:
1) The clustering of Q and K are done independently, in Line266 it says Cq=100 and Ck=500. Here lacks ablation study to clarify why Q/K clustering needs to be done independently, and how to determine the value of group numbers for each.
2) Line269, it says "We skip performing sparse attention on the first 30% denoising steps", but didn't mention how many steps are actually used in the experiments.
3) Line132~133, it says "13% of the computations are sufficient to achieve an attention recall of 95%", here needs clarification of the exact definitions of "computations" and "attention recall", and how the numbers are concluded.

---

> ### Author Rebuttal · Authors · 2025-07-31
>
> Dear Reviewer 8YJY,
>
> Thank you for your valuable questions. Below, we address each point raised.
>
> > Weakness 1: The clustering of Q and K are done independently, in Line266 it says Cq=100 and Ck=500. Here lacks ablation study to clarify why Q/K clustering needs to be done independently, and how to determine the value of group numbers for each.
>
> **Reply**: We conducted ablation experiments to test whether Q and K can share the same clustering strategy. Specifically, we evaluated three variants: applying Q clustering permutation ($\pi_Q$) to both Q and K, applying K clustering permutation ($\pi_K$) to both, and using clustering based on hidden states before QKV linear layer (i.e., shared QK embedding) for both sides ($\pi_S$). As shown in the table below, all three variants led to **worse PSNR** even with more computation budget (i.e., density) compared to clustering Q and K independently.
>
> | Permutation used by Q  | Permutation used by K  | Density  | PSNR    |
> |------------------------|-------------------------|----------|---------|
> | $\pi_Q$                | $\pi_K$                 | 31.28    | 26.562  |
> | $\pi_Q$                | $\pi_Q$                 | 38.23    | 22.439  |
> | $\pi_K$                | $\pi_K$                 | 38.58    | 22.183  |
> | $\pi_S$                | $\pi_S$                 | 87.27    | 26.495  |
>
> To further understand this, we compared the permutations of Q and K clustering and found that the permutation patterns differ substantially. Specifically, we calculate the Adjusted Rand Index (ARI) [1] value between Q clusters and K clusters, and the average ARI is 0.345, which is not very high. Therefore, clustering Q and K independently is necessary for preserving the expressiveness of attention.
>
> As for the number of clusters, we conducted ablations in the table below, varying the Q/K cluster counts and reported both PSNR and end-to-end efficiency. We found that setting Q = 100 and K = 500 strikes a **sweet balance** between quality and efficiency. While increasing the number of clusters improves the generation quality, the efficiency may decrease due to the hardware layout requirements. As tensor core requires a fixed layout (e.g., input size 64 for m64n64k16) to **saturate computation**, each Q cluster must contain more than 64 tokens on average. Cluster counts beyond Q=100 or K=500 begin to hurt efficiency due to underutilized computation.
>
> | QC  | KC   | PSNR   | SSIM  | LPIPS | Speedup  |
> |-----|------|--------|-------|-------|----------|
> | 100 | 250  | 25.497 | 0.801 | 0.182 | 1.90x    |
> | 100 | 1000 | 26.276 | 0.825 | 0.159 | 1.71x    |
> | 50  | 500  | 22.561 | 0.742 | 0.258 | 1.90x    |
> | 200 | 500  | 26.213 | 0.82  | 0.157 | 1.78x    |
> | 400 | 500  | 26.488 | 0.868 | 0.132 | 1.25x    |
> | 100 | 500  | 26.128 | 0.816 | 0.169 | 1.89x    |
>
>
> [1] https://en.wikipedia.org/wiki/Rand_index
>
>
> > Weakness 2: Line269, it says "We skip performing sparse attention on the first 30% denoising steps", but didn't mention how many steps are actually used in the experiments.
>
> **Reply**: Thank you for pointing this out. Our experiments use a total of 50 denoising steps, with the first 15 steps (30%) skipped for sparse attention as a warm-up phase.
>
> > Weakness 3: Line132~133, it says "13% of the computations are sufficient to achieve an attention recall of 95%", here needs clarification of the exact definitions of "computations" and "attention recall", and how the numbers are concluded.
>
> **Reply**: Thank you for the suggestion. The “13% of the computations” refers to **attention density**, i.e., the percentage of attention map retained in the sparse attention. “Attention recall” measures the proportion of the total attention score (after softmax) captured by the selected sparse tokens.
>
> In this case, retaining just 13% of token pairs preserves 95% of the total attention weight, indicating that our selection effectively focuses on the **most important attention regions**. We will revise the text to make these definitions explicit.
>
> > Question 1: How much gain can we get if the proposed method is applied to DiT based image generation model? This is not discussed in the manuscript.
>
> **Reply**: Thank you for raising this valuable point. We experimented with applying SAPAttn to the DiT-based image model Flux.1-dev, using a top-p threshold of 0.9 and a warmup ratio of 15%. We observed a PSNR of 25.3, indicating that SAPAttn preserves image generation quality.
> In terms of acceleration, SAPAttn achieved an end-to-end speedup of 1.05x. The gain is modest because Flux.1-dev operates at a context length of 4096, where attention accounts for only 15.6% of total runtime. As such, SAPAttn yields more substantial benefits in models with **longer context** lengths. This suggests that SAPAttn is particularly effective for video generation models or other settings where attention becomes a dominant computational bottleneck.
>
> > Question 2: The contribution of the work lies mostly in its custom implementation of "sparse attention", will the code be released? what's the schedule?
>
> **Reply**: Yes. We have implemented both FlashAttention v2 (for A100) and FlashAttention v3 (for H100) compatible versions of dynamic block sparse attention. This core component has already been open-sourced and integrated into the FlashInfer library. The remaining components of our system will also be open-sourced shortly to facilitate further research and practical adoption.

---

> > ### Author Response · Authors · 2025-08-06
> >
> > Thank you once again for your thoughtful suggestions and comments. As the discussion period deadline approaches, we would greatly appreciate it if you could let us know whether our responses have adequately addressed your concerns.
> >
> > Best,
> >
> > Authors

---

> ### Comment · Reviewer_8YJY · 2025-08-08
>
> Sorry for being late to respond, I've read the rebuttal from the authors and parallel reviews from other reviewers. All my questions are properly responded. I appreciate authors' honest and effort to disclose the acceleration result on image generation task,  the result is consistent with our recent validation. I'll update my final score.

---

> > ### Author Response · Authors · 2025-08-08
> >
> > Thank you for your considerate confirmation. We’re glad the revised results resolved your concerns, and we will reflect all changes in the camera-ready version.

---

### Official Review · Reviewer_31M2 · 2025-06-24

**Clarity:** 4
**Significance:** 3
**Originality:** 3
**Rating:** 5
**Confidence:** 3

**Summary:**

The paper presents a method for accelerating video generation with sparse attention via semantic-aware permutation. It details techniques such as k-means clustering for semantic-aware permutation and centroid-based top-p selection for critical token identification. Experiments show that its pipeline achieves significant speedup while maintaining high-quality generation results, outperforming other sota methods.

**Questions:**

1. In Table 1, why do the performance metrics (like PSNR) vary significantly across different models(Hunyuan vs Wan2.1)? What underlying principles explain these performance differences?
2. In the benchmarking results for the Hunyuan model in Table 1, the SVG[1]  also includes tests on the Hunyuan model. Specifically, the SVG+FP8 model achieves a speedup of 2.33x while maintaining similar quality metrics. This result appears comparable to the speedup metric of the author's model. Why did the authors choose not to include this result in this comparisons?
3. Building upon Question 1: The paper mentions "the lack of discussion and evaluation on whether the proposed methods can be extended to attention mechanisms other than DiTs." Given the noticeable differences in the method's effectiveness across models in Table 1, under what circumstances might the proposed methods fail? In other words, what are the capability boundaries and generality of the method?
4. In Section 4.3, regarding "customized attention kernels," the original text states, "By gathering blocks with various sizes on on-chip memory from global memory, SAPAttn is able to utilize the accelerators." The implementation details here remain unclear. Could the authors provide a detailed explanation of the implementation?
5. In the XAttention row of Table 1, there seems to be an extra "s" in the FLOPs column.


Reference:

[1] Sparse videogen: Accelerating video diffusion transformers with spatial-temporal sparsity. In ICML, 2025.

**Ethical Concerns:**

["NO or VERY MINOR ethics concerns only"]

**Final Justification:**

Given that the authors have addressed all my concerns and demonstrated the robustness of the proposed pipeline—even in its FP8 version—I am revising my score to 5, recommending acceptance.

**Limitations:**

Please refer to the Questions section.

**Paper Formatting Concerns:**

No formatting issues found in this paper.

**Quality:**

4

**Strengths And Weaknesses:**

Strengths:

1. Clear Structure and Readability. The paper is well-structured with a logical flow, making it easy to follow. The coherent organization of sections—from problem definition to methodology and experiments—facilitates smooth comprehension.
2. Methodology is simple yet effective. By leveraging the k-means++ algorithm for Semantic-Aware Permutation and combining it with Centroid-Based Top-P Selection for critical token identification, the approach balances simplicity and efficacy. The approach is straightforward without complex theories, but works really well—showing how simple, practical solutions can make a big difference.
3. It maintains high generation quality while delivering over 2×+ end-to-end speedup.

Weaknesses:

I do not find obvious weaknesses, but there are some questions. Please refer to the Questions section.

---

> ### Author Rebuttal · Authors · 2025-07-31
>
> Dear Reviewer 31M2,
>
> We thank the reviewer for the acknowledgment of our novel contributions and insightful questions. Below we respond to the questions.
>
> > Question 1: In Table 1, why do the performance metrics (like PSNR) vary significantly across different models(Hunyuan vs Wan2.1)? What underlying principles explain these performance differences?
>
> **Reply**: The performance variation (e.g., PSNR differences) across models ( Hunyuan and Wan2.1) arises from inherent numerical stability. In our experiments, we found that Hunyuan is relatively robust against precision variance while Wan2.1 is highly sensitive.
> For instance, when evaluating the same dense attention using different backends (FlexAttention, FlashAttention, Torch SDPA), Wan2.1 exhibited PSNR as low as 27–28. However, HunyuanVideo exhibits ~33-34 PSNR despite no setup changes.
>
> Therefore, it is natural that SAPAttn achieves a lower PSNR on Wan compared with HunyuanVideo, due to its **sensitivity to numerical changes**. These differences are largely model-specific and reflect varying sensitivity to low-level numerical behaviors, which do not correlate with the performance of the methodology.
>
> > Question 2: In the benchmarking results for the Hunyuan model in Table 1, the SVG[1] also includes tests on the Hunyuan model. Specifically, the SVG+FP8 model achieves a speedup of 2.33x while maintaining similar quality metrics. This result appears comparable to the speedup metric of the author's model. Why did the authors choose not to include this result in these comparisons?
>
> **Reply**: We apply FP8 quantization in combination with SAPAttn and evaluate its impact on the Wan 2.1 I2V model. Compared with the version without FP8, our approach achieves a significant speedup with nearly lossless quality.
>
> |                | PSNR    | LPIPS  | Speedup  |
> |----------------|---------|--------|----------|
> | SAPAttn        | 26.562  | 0.138  | 1.58x    |
> | SAPAttn + FP8  | 26.545  | 0.141  | 1.76x    |
>
>
> > Question 3: Building upon Question 1: The paper mentions "the lack of discussion and evaluation on whether the proposed methods can be extended to attention mechanisms other than DiTs." Given the noticeable differences in the method's effectiveness across models in Table 1, under what circumstances might the proposed methods fail? In other words, what are the capability boundaries and generality of the method?
>
> **Reply**: SAPAttn primarily accelerates attention computation, which dominates runtime in long-context scenarios due to its quadratic complexity. Its benefits are thus most pronounced when **context length is large**.
> In addition, SAPAttn introduces a centroid cache to accelerate the k-means clustering process. which is especially effective in **iterative generation** tasks (e.g., diffusion or auto-regressive decoding) where context is reused across steps. In these cases, the centroid cache amortizes the clustering cost and contributes substantially to speedup. In settings where the context is used only once (e.g., prefill-only inference in LLMs), the lack of reuse limits the performance gains.
> These observations outline the capability boundaries of SAPAttn: **it offers the greatest benefits in long-context, multi-forward generation tasks where attention dominates the runtime and centroid reuse is possible.**
>
>
> > Question 4: In Section 4.3, regarding "customized attention kernels," the original text states, "By gathering blocks with various sizes on on-chip memory from global memory, SAPAttn is able to utilize the accelerators." The implementation details here remain unclear. Could the authors provide a detailed explanation of the implementation?
>
> **Reply**: Our sparse attention with variable block sizes is built on FlashAttention3 (FA3), combining sparse loading and dense computation. To maximize hardware efficiency, SAPAttn uses the `wgmma` with an `m64n64k16` shape. For query tokens, we load contiguous tokens from the same cluster, which are naturally contiguous in memory after permutation. For key/value tokens, which may be scattered in global memory due to varying cluster sizes, SAPAttn uses per-token address offsets to perform sparse loading and stores them in shared memory in a contiguous layout. This enables efficient use of MMA instructions without the need for expensive key/value padding, leading to over 85% of the theoretical maximum performance, where the upper bound is estimated by multiplying the sparsity density with the runtime of the dense FlashAttention-3.
>
> > Question 5: In the XAttention row of Table 1, there seems to be an extra "s" in the FLOPs column.
>
> **Reply**: Thanks for pointing that out. We will remove it in our camera-ready version.

---

> > ### Comment · Reviewer_31M2 · 2025-08-02
> > **Re:Rebuttel**
> >
> > I thank the authors for the rebuttal, which has addressed most of my concerns, except for one point:
> > In their reply to Q2, the authors provide an ablation study of FP8 for the Wan 2.1 I2V model to demonstrate that the proposed SAPAttn, after FP8 quantization, can further improve speed without losing quality. However, my question is that in Table 1 of the SVG (Please refer to the SVG paper), the SVG+FP8 setting shows performance in both quality and efficiency that is roughly comparable to the authors' proposed SAPAttn (with both being test results on the Hunyuan T2V). Therefore, I suggest that the authors further supplement the test results of SAPAttn + FP8 on Hunyuan T2V to prove the superiority of their method over SVG. (Authors have already included comparison with SVG but without FP8 in Table 1, main paper)

---

> ### Author Response · Authors · 2025-08-03
> **SAPAttn with FP8 on Hunyuan**
>
> We thank the reviewer for this thoughtful follow-up! In response, we conducted an additional experiment on the Hunyuan T2V model to evaluate the impact of FP8 quantization on SAPAttn. We used the exact same set of evaluation videos and random seeds.
>
> As shown in the table below, SAPAttn and SAPAttn+FP8 outperforms SVG and SVG+FP8 in both quality and efficiency. We greatly appreciate your feedback and will include these results in the revised manuscript.
>
> | Method         | PSNR   | SSIM   | LPIPS  | Speedup |
> |----------------|--------|--------|--------|---------|
> | SVG            | 29.157 | 0.905  | 0.120  | 1.91×   |
> | SVG + FP8      | 29.033 | 0.902  | 0.121  | 2.33×   |
> | SAPAttn        | 30.452 | 0.910  | 0.117  | 2.30×   |
> | SAPAttn + FP8  | 30.389 | 0.908  | 0.118  | 2.55×   |

---

> > ### Comment · Reviewer_31M2 · 2025-08-05
> > **My coconers have been adressed.**
> >
> > Thanks the authors, my concerns have been fully addressed.

---

> ### Author Response · Authors · 2025-08-05
>
> Thank you for confirming that your concerns have been fully addressed. We would greatly appreciate it if you could consider updating your evaluation to reflect your current view of the paper.
>
> Best,
>
> Authors

---

### Official Review · Reviewer_PvVz · 2025-06-30

**Clarity:** 2
**Significance:** 3
**Originality:** 2
**Rating:** 4
**Confidence:** 4

**Summary:**

This paper proposes a sparse attention via semantic-aware permutation to accelerate video diffusion transformers, which clusters and reorders tokens based on semantic similarity using k-means, providing cluster representations for efficient computation. However, the demonstration of this paper is poor, which makes it confusing about how it achieves this goal, including the implementation details about k-means and the attention mechanism based on cluster tokens.

**Questions:**

The implementation details can be refined to facilitate the understanding of the proposed method.

**Ethical Concerns:**

["NO or VERY MINOR ethics concerns only"]

**Final Justification:**

My coconers have been fully adressed.

**Limitations:**

yes

**Quality:**

2

**Strengths And Weaknesses:**

**Strengths:**
- The proposed method is training-free and can be seamlessly transferred to pre-trained DIT-based video diffusion models.
- From the provided quantitative results, the proposed method achieves better content preservation under the same speedup.

**Weaknesses:**
- The paper claims that it employs k-means for token clustering and uses a centroid cache for fast convergence. However, it is unclear how to determine the optimal number of centroids, as the number of areas with different semantics can vary significantly in different videos. A limited number of centroids may cluster tokens with different semantics together, while a large number of centroids may limit acceleration performance.
- It claims that it permutes tokens within each cluster into a contiguous layout for efficient computation. Does this technique only support the proposed method? It would be beneficial to transfer this technique to other methods and verify its performance in the same setting. This technique seems to be more of an engineering trick.
- How is sparse attention performed based on the clustered tokens? Is it through token merging or attention only within the same cluster?
- How does the top-k approach ensure semantic-aware clustering? There is no visualization of the clustered tokens using top-k.

---

> ### Author Rebuttal · Authors · 2025-07-31
>
> Dear Reviewer PvVz,
>
> Thanks for your insightful reviews. Below, we provide point-by-point responses to each of your comments.
>
> > Weakness 1: The paper claims that it employs k-means for token clustering and uses a centroid cache for fast convergence. However, it is unclear how to determine the optimal number of centroids, as the number of areas with different semantics can vary significantly in different videos. A limited number of centroids may cluster tokens with different semantics together, while a large number of centroids may limit acceleration performance.
>
> **Reply**: Thanks for the insightful question. In our response below, we use QC to denote the number of query centroids and KC to denote the number of key centroids. We empirically select KC and QC by a comprehensive ablation study of both efficiency and accuracy.
>
> **Rationale for the choice of QC:**
> With KC fixed at 500, we analyze the effect of varying QC. As shown in Table 1, increasing QC from 50 to 100 yields a notable improvement in generation quality. However, further increasing QC to 200 or 400 leads to a sharp drop in computational efficiency. This degradation stems from constraints in the sparse attention kernel due to the hardware layout requirements, where the tensor core requires a fixed layout (e.g., input size 64 for m64n64k16) to saturate computation. As shown in Table 2, we observe a clear latency increase when QC > 100, and therefore, we choose QC = 100 as the optimal balance point.
>
>
> **Rationale for the choice of the KC:**
> With QC fixed at 100, we investigate the impact of varying KC. Compared to KC = 250, KC = 500 yields significantly improved visual quality while only being slightly slower. Compared to KC = 1000, though KC = 1000 achieves slightly higher visual quality, KC = 500 is faster in terms of end-to-end latency. Both of them lie on the Pareto frontier with respect to the trade-off between generation quality and computational efficiency. In our case, we select KC = 500 for efficiency.
>
> Table 1: Generation quality and speedup when varying QC and KC.
> | QC  | KC   | PSNR   | SSIM  | LPIPS | Speedup  |
> |-----|------|--------|-------|-------|----------|
> | 100 | 250  | 25.497 | 0.801 | 0.182 | 1.90x    |
> | 100 | 1000 | 26.276 | 0.825 | 0.159 | 1.71x    |
> | 50  | 500  | 22.561 | 0.742 | 0.258 | 1.90x    |
> | 200 | 500  | 26.213 | 0.82  | 0.157 | 1.78x    |
> | 400 | 500  | 26.488 | 0.868 | 0.132 | 1.25x    |
> | **100** | **500**  | **26.128** | **0.816** | **0.169** | **1.89x**    |
>
>
> Table 2: Sparse attention kernel's latency when varying QC.
>  Latency (ms)   | QC = 50 | QC = 100 | QC = 200 | QC = 400  |
> |----------------|---------|----------|----------|-----------|
> | Sparsity = 0.3 | 69.48   | 70.86    | 79.3     | 107.95    |
> | Sparsity = 0.1 | 23.49   | 24.4     | 28.59    | 34.7      |
>
>
>
> > Weakness 2: It claims that it permutes tokens within each cluster into a contiguous layout for efficient computation. Does this technique only support the proposed method? It would be beneficial to transfer this technique to other methods and verify its performance in the same setting. This technique seems to be more of an engineering trick.
>
> Reply: We respectfully disagree with the opinion that dynamic permutation is an engineering trick. Instead, it is a core component of **an algorithm-hardware co-design** that is tightly integrated with our token-level kmeans identification. Our key insight is that critical tokens can be **highly scattered** across the attention map, which fundamentally motivates the need for permutation. This scattering only occurs with token-level identification, and reordering these tokens into contiguous layouts is necessary for efficient execution on modern hardware.
>
> In contrast, prior block-level identification methods[1-2] (e.g., mean or max pooling) inherently produce block-contiguous layouts but suffer from coarse, inaccurate selection and significant computation waste. These methods do not benefit from permutation, as their sparsity patterns are already aligned for compute. Our approach, by contrast, leverages token-level semantic clustering via k-means to enable dynamic permutation and unlock substantial speedups. Thus, permutation is not a general-purpose trick but a mechanism **deeply co-designed** with SAPAttn’s core method.
>
> [1] Xu, Ruyi, et al. "Xattention: Block sparse attention with antidiagonal scoring." arXiv preprint arXiv:2503.16428 (2025).
>
> [2] Zhang, Jintao, et al. "Spargeattn: Accurate sparse attention accelerating any model inference." arXiv preprint arXiv:2502.18137 (2025).
>
> > Weakness 3: How is sparse attention performed based on the clustered tokens? Is it through token merging or attention only within the same cluster?
>
> Reply: We would like to clarify that SAPAttn neither merges tokens nor performs attention only within the same cluster. In SAPAttn, query and key tokens are clustered independently using k-means. Based on the centroids of clusters, we get an estimation of the attention map and apply top-p selection for the estimated attention scores to determine, for each query cluster, which key clusters are critical. For each selected query-key cluster pair, every query token in the query cluster attends to all key tokens in the corresponding key cluster.
>
> > Weakness 4: How does the top-k approach ensure semantic-aware clustering? There is no visualization of the clustered tokens using top-k.
>
> **Reply**: Thank you for the comment. We would like to clarify that our method uses a top-p selection strategy, where the number of critical tokens is determined dynamically based on their importance, rather than using a fixed budget as in top-k selection. This allows for dynamic, content-adaptive selection of critical tokens based on cumulative attention scores. In Figure 6 (b), we provide a visualization of the attention map after semantic-aware permutation, where the tokens are clustered together through permutation. This clearly shows that semantically similar tokens are grouped together, making the attention map very condensed, thus being hardware-friendly.
>
> > Question 1: The implementation details can be refined to facilitate the understanding of the proposed method.
>
> **Reply**: Thank you for the suggestion. To provide a comprehensive clarification, we summarize the algorithmic workflow and our implementation below:
>
> **Algorithm-level clarification:**
>
> We summarize the key steps of SAPAttn as follows:
>
> 1. Independent Q/K clustering:
> Query and key tokens are clustered separately using k-means into 100 and 500 clusters, respectively. To accelerate convergence, we use the centroids from the previous diffusion step as initialization. After clustering, tokens within each query/key cluster are permuted to be contiguous in memory.
>
>
> 2. Centroid-level attention estimation and top-p selection:
> Each query cluster estimates attention scores to all key clusters via centroid-to-centroid computation, weighted by key cluster sizes. The top key clusters covering a cumulative top-p mass are selected, forming a sparse attention pattern.
>
>
> 3. Sparse attention execution:
> Each token attends only to tokens in the selected key clusters, enabling token-level sparsity while preserving semantic coverage.
>
>
> **Implementation-level clarification:**
>
> Our dynamic block-sparse attention kernel supports both FA2 (A100) and FA3 (H100). For FA3, we use wgmma (m64n64k16) for dense compute, with sparse per-token key/value loading via offset indexing and rematerialization in shared memory—achieving high throughput without padding overhead.

---

> > ### Comment · Reviewer_PvVz · 2025-08-03
> > **My coconers have been adressed.**
> >
> > My concerns regarding the implementation details have been fully addressed, and it is suggested to revise the manuscript in accordance with the rebuttal, incorporating the detailed cluster strategy and additional visualizations of the semantic-aware clustering.

---

> ### Author Response · Authors · 2025-08-04
>
> Thank you for your thoughtful confirmation. We're glad the new results addressed your concerns, and will update all contents in the camera-ready version.

---

### Official Review · Reviewer_QGvt · 2025-07-03

**Clarity:** 3
**Significance:** 3
**Originality:** 3
**Rating:** 4
**Confidence:** 5

**Summary:**

This paper introduces SAPAttn, a training-free sparse attention framework designed to accelerate video generation based on Diffusion Transformers (DiTs). The paper's core contribution lies in its semantic-aware permutation, top-p dynamic budget control, and customized kernel implementations, which together achieve a Pareto frontier trade-off between generation quality and efficiency.

**Questions:**

see weakness

**Ethical Concerns:**

["NO or VERY MINOR ethics concerns only"]

**Final Justification:**

My concerns regarding the details have been fully addressed.

**Limitations:**

see weakness

**Quality:**

3

**Strengths And Weaknesses:**

Strengths
The proposal of semantic-aware permutation is novel. By clustering tokens based on semantic similarity using k-means and reordering them, the method improves the identification accuracy of critical tokens and reduces computation waste.


Weaknesses and Suggestions
1. In Figure 5(b), some key centroids are shown in both red and blue. The authors need to explain the rationale behind these cross-category centroids. Are they determined based on specific semantic associations, or is it a random occurrence? A detailed theoretical analysis or experimental verification should be provided to address this confusion.
2. The paper should explicitly clarify whether critical tokens are equivalent to cluster centers. This would help readers better understand the core idea of the proposed method.
3. In Table 1, HunyuanVideo achieves a 2.30× speedup at a density of 25.45%, while Wanx only achieves a 1.89× speedup at a density of 12.87%. The authors should provide a detailed analysis of the reasons for this difference, considering factors such as computational efficiency and model architecture differences.
4. The paper mentions using VBench for evaluation but does not test the main benchmark metrics of VBench. These main benchmark metrics are crucial for comparing the performance of text-to-video models. The lack of such tests makes it difficult to fully assess the performance advantages of SAPAttn. The authors should supplement the evaluation with VBench main benchmark metrics and report the results in detail.
5. As far as I know, Flash Attention does not support attention masks and only supports the original full attention mechanism. The complex attention mechanisms proposed in this paper may not be compatible with Flash Attention v3. However, the paper only provides FLOPs comparisons instead of direct time comparisons, which may limit the assessment of acceleration effects. The authors should consider using Flash Attention v3 as the baseline and directly comparing the total time for video generation to more accurately reflect the actual acceleration performance of SAPAttn.
6. However, the completeness and usability of the code and data cannot be verified during the review process. The authors should provide the code and data, for example, detailed full data of each video in VBench evaluation to the reviewer, and usable after release to facilitate further research by other researchers.

---

> ### Author Rebuttal · Authors · 2025-07-31
>
> Dear Reviewer QGvt,
>
> Thanks for your insightful reviews. Below, we provide point-by-point responses to each of your comments.
>
> > Weakness 1: In Figure 5(b), some key centroids are shown in both red and blue. The authors need to explain the rationale behind these cross-category centroids. Are they determined based on specific semantic associations, or is it a random occurrence? A detailed theoretical analysis or experimental verification should be provided to address this confusion.
>
> **Reply**: The appearance of shared key centroids is an expected outcome of sparse attention. In SAPAttn, query and key tokens are **clustered independently**, where each query cluster selects its key/value clusters based on centroid-level attention scores. Since this selection is done independently for each query cluster, it is natural for the same key cluster to be selected by multiple query clusters, forming a many-to-many mapping. For example,  one key/value cluster that encodes semantically important information can be selected by all the other query clusters. This behavior is clearly shown in Figure 6, where certain key clusters are densely attended by multiple query clusters in the “after permutation” attention maps.
>
> This phenomenon is also well observed by attention sink [1] in LLMs, where the first key/value tokens are highly attended to by most query tokens.  Therefore, such flexible many-to-many mapping widely exists and is important for high-quality generation, which is inherently supported by SAPAttn’s design.
>
> [1] Xiao, Guangxuan, et al. "Efficient streaming language models with attention sinks." arXiv preprint arXiv:2309.17453 (2023).
>
> > Weakness 2: The paper should explicitly clarify whether critical tokens are equivalent to cluster centers. This would help readers better understand the core idea of the proposed method.
>
> **Reply**: Thanks for the suggestion, and we will incorporate this discussion into the revised manuscript.
>
> We respectfully clarify that critical tokens and cluster centers serve different roles in our method. Cluster centers are computed during semantic-aware permutation via k-means and act as representatives of semantically similar token clusters. In contrast, critical tokens are dynamically selected based on attention scores, and can be approximated by a subset of cluster centers.
>
> While critical tokens often reside within high-attention clusters, they are not equivalent to the whole cluster centers — they form a subset guided by the importance derived from centroid-based estimation.
>
> > Weakness 3: In Table 1, HunyuanVideo achieves a 2.30× speedup at a density of 25.45%, while Wanx only achieves a 1.89× speedup at a density of 12.87%. The authors should provide a detailed analysis of the reasons for this difference, considering factors such as computational efficiency and model architecture differences.
>
>
> **Reply**: Thank you for the valuable comment. The difference in end-to-end speedup between HunyuanVideo and Wan primarily stems from their varying attention cost ratios, which are mainly due to different context lengths and model architectures.
>
> Specifically, HunyuanVideo’s context length is 118k, while Wanx‘s context length is 75k. HunyuanVideo has 2 parts in its layers: Self Attention and Feed-Forward Network, while Wanx has an additional cross attention block.
>
> Therefore, the attention proportion in HunyuanVideo will be larger than WanX. Since our method primarily accelerates the attention module via SAP, the overall speedup naturally scales with its contribution to total runtime. We will revise Table 1 to separate attention-level and end-to-end speedups to improve clarity explicitly.
>
>
> > Weakness 4: The paper mentions using VBench for evaluation but does not test the main benchmark metrics of VBench. These main benchmark metrics are crucial for comparing the performance of text-to-video models. The lack of such tests makes it difficult to fully assess the performance advantages of SAPAttn. The authors should supplement the evaluation with VBench main benchmark metrics and report the results in detail.
>
> **Reply**: We evaluate VBench main benchmark metrics on Wan 2.1 T2V and HunyuanVideo T2V models. We find that SAPAttn consistently performs better than baselines on VBench, demonstrating the effectiveness of our method.
>
> |    Wan 2.1     | SubConsis | BackConsis | MotionSmooth | AesQual | ImagQual  |
> |---------|-----------|------------|--------------|---------|-----------|
> | Dense   | 0.956     | 0.968      | 0.983        | 0.613   | 0.713     |
> | Sparge  | 0.927     | 0.948      | 0.978        | 0.567   | 0.684     |
> | SVG    | 0.947     | 0.96       | 0.98         | 0.597   | 0.703     |
> | SAPAttn | 0.954     | 0.965      | 0.982        | 0.602   | 0.709     |
>
>
> |   HunyuanVideo      | SubConsis | BackConsis | MotionSmooth | AesQual | ImagQual  |
> |---------|-----------|------------|--------------|---------|-----------|
> | Dense   | 0.915     | 0.941      | 0.993        | 0.648   | 0.753     |
> | XAttn   | 0.912     | 0.924      | 0.992        | 0.631   | 0.739     |
> | SVG    | 0.914     | 0.928      | 0.993        | 0.652   | 0.739     |
> | SAPAttn | 0.917     | 0.946      | 0.993        | 0.657   | 0.751     |
>
>
> > Weakness 5: As far as I know, Flash Attention does not support attention masks and only supports the original full attention mechanism. The complex attention mechanisms proposed in this paper may not be compatible with Flash Attention v3. However, the paper only provides FLOPs comparisons instead of direct time comparisons, which may limit the assessment of acceleration effects. The authors should consider using Flash Attention v3 as the baseline and directly comparing the total time for video generation to more accurately reflect the actual acceleration performance of SAPAttn.
>
> **Reply**: We appreciate the reviewer's valuable comment on the compatibility of SAPAttn with Flash Attention v3 (FA3). We want to clarify that SAPAttn is fully compatible with FA3. Specifically, we implemented SAPAttn in both FA2 and FA3 versions, ensuring substantial speedups on both A100 and H100 GPUs. The table below shows the latency under the Wan 2.1 setting with sequence length 74256 on H100, comparing sparse and dense attention using both FA2 and FA3 across various densities.
>
> | Density      | 0.1  | 0.2  | 0.3   | 0.4   | 0.5   | 1.0 (original FA2/FA3)  |
> |--------------|------|------|-------|-------|-------|-------------------------|
> | SAP-FA2 (ms) | 35.5 | 71.2 | 107.1 | 141.3 | 176.9 | 317.7                   |
> | SAP-FA3 (ms) | 24.4 | 47.6 | 70.9  | 95.4  | 117.7 | 195.6                   |
>
> > Weakness 6: However, the completeness and usability of the code and data cannot be verified during the review process. The authors should provide the code and data, for example, detailed full data of each video in VBench evaluation to the reviewer, and usable after release to facilitate further research by other researchers.
>
> **Reply**: Due to the policy, we are unable to share anonymous links during the review process. However, the core component—dynamic block sparse attention—has already been open-sourced and integrated into the FlashInfer library. The remaining parts of the system, including code and evaluation data (e.g., full video metrics for VBench), will be released shortly to support both research reproducibility and practical adoption.

---

> > ### Comment · Reviewer_QGvt · 2025-08-04
> >
> > My concerns regarding the details have been fully addressed, and I will update my score.

---

> > > ### Author Response · Authors · 2025-08-04
> > >
> > > We appreciate your feedback and the time you’ve taken to review our work. We will include all the answers in the final manuscript.

---

> > > > ### Comment · Reviewer_QGvt · 2025-08-08
> > > >
> > > > Since this paper presents an in-depth study of acceleration/linear attention for video generation primarily from a non-pretraining perspective, it is recommended that the authors also cite recent works on acceleration/linear attention for video generation from the pretraining perspective in the related work section, such as:
> > > >
> > > > [1] "Matten: Video generation with mamba-attention." (2024).
> > > >
> > > > [2] "Lingen: Towards high-resolution minute-length text-to-video generation with linear computational complexity." (2025).
> > > >
> > > > [3] "M4V: Multi-Modal Mamba for Text-to-Video Generation." (2025).
> > > >
> > > > [4] "Long-context state-space video world models." (2025).

---

> > > > > ### Author Response · Authors · 2025-08-08
> > > > >
> > > > > Thank you for your suggestion. In the camera-ready version, we will cite the mentioned papers about linear attention / state-space models for video generation from the pretraining perspective.

---

### Comment · Area_Chair_3ccD · 2025-08-01
**Author-Reviewer Discussions (July 31 - Aug 6)**

Dear reviewers,

Authors have provided their rebuttals and now we are in Author-Reviewer Discussions (July 31 - Aug 6) period. Please read authors' rebuttal and give comments. Thanks!

---

### Decision · Program_Chairs · 2025-09-17

**Decision:**

Accept (spotlight)

**Comment:**

This paper proposed SAPAttn, a training-free framework that maximizes identification accuracy and minimizes computation waste, achieving a Pareto frontier trade-off between generation quality and efficiency for video diffusion transformers.

Paper got 2 Borderline accept, 2 Accept rating.

Before rebuttal, reviewers thought the strengths of this paper are:
1) method is novel (Reviewer QGvt, 8YJY )
2) method shows improvement. (Reviewer QGvt, PvVz, 31M2, 8YJY)
3) proposed method is training-free. (Reviewer PvVz)
4) paper is well written (Reviewer 31M2, 8YJY)

Weaknesses are:
1) need more experiments (Reviewer QGvt)
2) more clarification and reason for some of the design and experiments (Reviewer QGvt, PvVz, 31M2, 8YJY)

After the rebuttal, all the reviewers thought authors addressed their concerns and gave accept or borderline accept rating. Given these AC decided to accept this paper.